# The Development of ASIC Type GSR Sensor Driven by GHz Pulse Current [note 1]

**DOI:** 10.3390/s20041023

**Published:** 2020-02-14

**Authors:** Yoshinobu Honkura, Shinpei Honkura

**Affiliations:** 1Magnedesign Corporation, Nagoya 466-0059, Japan; 2Nanocoil Corporation, Nagoya 466-0059, Japan

**Keywords:** GSR sensor, GMI sensor, GHz pulse, magnetic sensor, amorphous wire

## Abstract

The GigaHertz spin rotation (GSR) effect was observed through the excitement of Giga Hertz (GHz) pulse current flowing through amorphous wire. The GSR sensor that was developed provides excellent features that enhanced magnetic sensitivity and sine functional relationship, as well as good linearity, absence of hysteresis, and low noise. Considering the GHz frequency range used for the GSR sensor, we assume that the physical phenomena associated with the operation of the sensor are based on spin reduction and rotation of the magnetization. The proper production technology needed was developed and a micro-sized GSR sensor was produced by directly forming micro coils on the surface of the application-specific integrated circuit (ASIC). Some prototypes of the ASIC type GSR sensor have been produced in consideration of applications such as automotive use, mobile device use, and medical use. Therefore, we can conclude that GSR sensors have great potential to become promising magnetic sensors for many applications.

## 1. Introduction

Magnetic sensors have been used for the monitoring of various processes and functions in a great number of industries such as data storage, home entertainment, energy harvesting and conversion, informatics, telecommunication, aircraft, aerospace, automobile, electronic surveillance, medicine, biology, etc., over many decades [1,2,3].

There are several physical phenomena that allow the measurement of magnetic fields, and most of them are based on the use of different magnetic materials.

Particularly, soft magnetic materials serve as the basic material for a wide scale of different sensors [3].

Amorphous and nanocrystalline materials prepared by rapid melt quenching present a number of relevant advantages such as excellent magnetic softness, a fast and inexpensive fabrication process, dimensionality suitable for various sensor applications and in certain cases, biocompatibility [4,5,6].

Amorphous and nanocrystalline ferromagnetic wires can exhibit very peculiar properties suitable for magnetic sensor applications, such as giant magnetoimpedance (GMI), or single domain wall propagation [6,7,8]. Notably, the above mentioned GMI was the first reported crystalline magnetic wire [9].

Studies of the GMI effect attracted the attention of researchers and engineers in recent years owing to excellent impedance sensitivity of external magnetic field (up to 10%/A/m): the largest one among non-cryogenic effects [10,11,12]. Up to now the highest GMI ratio (up to 650%) is reported for properly prepared and processed amorphous microwires [13,14,15].

Consequently, the GMI technology has been proposed for numerous applications, like magnetic compasses and acceleration sensors integrated in complementary metal oxide semi-conductor (CMOS) circuits [16,17], low-sized magnetometers suitable for magnetic field mapping [17,18,19], and the detection of bio magnetic fields with the pico-Tesla sensitivity [20] or magneto elastic sensors [21]. Amorphous wires are used for all reported sensors. 

It is widely accepted that the origin of the GMI effect is related to the skin effect of magnetically soft conductors [7,8,9,10,11,12]. However, for high enough frequencies, the domain walls are strongly dampened. In fact, the employment of thinner soft magnetic wires required for miniaturization of the sensors and devices requires an extension of the frequency range for the impedance toward the higher frequencies (GigaHertz (GHz) range) [22,23]. Moreover, GHz frequency impedance changes induced by external magnetic field have been attributed to ferromagnetic resonance (FMR) [22]. Therefore, spin precession and magnetization rotation on the surface layer of magnetic wires must be considered when interpreting the impedance change upon external magnetic field.

Given the attractive magnetic properties of amorphous wires, the development of magnetic sensors using magnetic wires has aroused great interest in recent years.

Highly sensitive micro magnetic sensors based on amorphous wire, such as FluxGate sensor (FG sensor) [24] introduced in 1987, GMI sensor [25] introduced in 1999, and GSR sensor [26] introduced in 2015 have shown progress in both sensitivity and size for extending Internet of Things (IoT) applications. The FG sensor and the GMI sensor have been used for electronic compasses [24,25,26,27] for automotive and smartphone uses [28,29]. These sensors are based on the same principle of allowing the measure of the voltage (Vc) of the coil surrounding the amorphous wire proportional to the external magnetic field on passing pulse current through the wire. As a result, they have the same element structures of these sensors and the same design of these electronic circuits.

The sensitivity is expected in principle to be dependent on Vc ∞ √ f・N・μH, where *f* is the frequency, N is the number of coil turns, and μH is the wire’s permeability. The pulse current frequency of FG sensor, GMI sensor, and GSR sensor has been increased from KiloHertz (KHz) and MegaHertz (MHz) [30], and to GHz range. Consequently, the physics involved in GSR sensor and the features are different from those involving GMI effect; GSR sensors present larger sensitivity per element volume.

The different frequencies employed for GMI cause different electro-magnetic phenomena, even if the same amorphous wires are used. FG sensors detect the rotation of axial magnetization generated in whole cross section of the wire induced by KHz pulse current. GMI sensor principle is based on the skin effect in magnetic conductors induced by MHz pulse current, which detects the rotation of the axial magnetization of the skin layer of the wire, which is related to the domain wall movement inside the magnetic wall [30]. The GSR sensor working at GHz range caused a new phenomenon, involving the magnetization rotation on the surface with tilt angle toward axis direction dependent on the magnetic field which is arisen by GHz pulse current. This resulted in the creation of a sine functional output relationship between the coil voltage and the magnetic field as well as good linearity, no hysteresis, and low noise. This new phenomenon was named as GSR effect.

The sensitivity of the GSR sensor increases with the increase of the number of coil turns per length, which can be made by 3-dimensional photolithography technique [30] allowing to produce fine pitched micro coil. The micro coil with the coil pitch of 5.5 micrometers (μm) and diameter of 16 μm is produced by this new technique. Current GMI sensor have a coil pitch of 30 μm and a diameter of 32 μm. The new coil has a six times smaller coil pitch than that of conventional coil. At the same time the micro coil is produced directly on the protective film of ASIC surface [30] so that this new process allows the development of a micro sized ASIC type GSR sensor without the assembling process of ASIC and the element.

In this paper we present the studied effects of the GSR sensor, and its many features associated with the GHz pulse frequency on coil voltage, the relationship between the coil voltage and the external field, and hysteresis and noise properties of ASIC type GSR sensor. Subsequently, the effect of new process to produce micro coil and the sensor’s possibility for various applications.

## 2. Research on GSR Effect

### 2.1. Principle of GSR Effect

The principle of the GSR effect induced in the amorphous wire with zero magnetostriction is explained in Figure 1. The wire has a special magnetic domain structure [31] that consists of surface domains with circular spins, axis magnetic domains, and 90-degree domain wall existing between the two domains. When an external magnetic field is applied to the wire along axis direction, the electronic spins in the surface domain tilt toward the axial direction, with the angle dependent on the magnetic field strength. The axis magnetic domains induce axial direction magnetization. The GHz pulse current passes through the wire to make a strong circular magnetic field and only make spin rotation with GHz angular velocity but not the movement of the domain wall, because of strong skin effect induced by GHz current pulse. Figure 2 shows the typical planed structure of GSR element, consisting of one glass coated amorphous wire, two wire electrodes, and two coil electrodes with the length of 0.16 mm and width of 0.23 mm. Figure 3 shows an observed result of the wire voltage and the coil voltage induced by GHz pulse current. The peak coil voltage is induced on the sharp edges of the rise and fall of the wire pulse current. It is predicted that: (1) GSR effect may increase the coil voltage with the increase in frequency of GHz pulse current and number of coil turns, and (2) the spin rotation not accompanied with domain wall movements could improve the magnetic properties of magnetic noise, hysteresis, and linearity, as well as sensitivity

### 2.2. Experimental Procedure

The present GSR element shown in Figure 2 equips a wire with a composition of Co_50.7_Fe_8.1_B_13.3_Si_10.3_ [31] and the permeability of 1800 with a diameter of 10 μm. The tested GSR elements have lengths of 0.16 mm, 0.45 mm, and 0.96 mm, a wire resistance of 3 Ω, 4.5 Ω, 8 Ω, and 13 Ω, coil turn numbers of 14, 32, 66, and 148, and finally a coil resistance of 80 Ω, 210 Ω, 360 Ω, and 810 Ω respectively.

The block diagram and ASIC of electronic circuit for GSR sensor in Figure 4a is similar to the conventional GMI circuit [25,26,32] but ASIC used in this research has improvements as follows. The pulse generator can generate pulse currents with frequency of 1 GHz to 3 GHz. The electronic switch can operate at a very small interval of 0.1 ns between on and off. The adjustment circuit can control the detection timing from 0 to 4 ns by interval of 0.1 ns. The analog circuit has band width of 500 KHz and AD converter has 16 bits. The I2C communication is used to send data to MCU. The consumption current is about 0.4 mA @ ODR of 5 KHz.

Our experiments show that connecting the ASIC and GSR elements by wire bonding (Figure 4b produces effects in pulse frequency, detection timing, coil turn numbers, and effective permeability on magnetic properties such as sensitivity, relationship between magnetic field and coil voltage, measuring range, linearity, noise, and hysteresis. The effect of frequency is examined by changing the transition time of the pulse current Δt from 0.2 ns to 1 ns where pulse frequency f is defined by f = 1/2 Δt.

### 2.3. Results on Features of Coil Voltage of GSR Sensor

The coil voltage of the GSR sensor [26,33] observed under a frequency of 1.5 GHz reaches a maximum value of about 1 ns and then decreases. The amplitude of the coil voltage increases as the strength of the magnetic field increases and the sign of the amplitude takes positive and negative values according to positive and negative value of the magnetic field. It is noted that coil voltages at H = 0 A/m are very small compared to those at H = 720 A/m, which means magnetic signal voltage. A relationship between the coil voltage and the magnetic field at the maximum detection timing of the falling process is shown in Figure 5. There is a surprising result that the relationship has a sine function expressed by the equation V = V_0_・sin(πH/2Hm) where Hm is the field strength taking Vmax. The experimental data results show that Hm is equal to the anisotropy Hk of the amorphous wire, that is, Hm = 0.96Hk.

Figure 6 shows both linear lines of an inverse voltage by πH/2Hm = arcsin(V/V_0_)and a regression line. It is found that the linear relationship between the coil voltage and the magnetic field gives good linearity of 0.5% FS and an extension of the measuring range from 960 A/m (linear approximation) to 7200 A/m (dependent on Hm). On the contrary, the GMI sensor output is based on the BH curve (magnetic flux density (B) × magnetic field strength (H)) of amorphous wire without a mathematical equation so that the collinear approximation would not be used to extend the measuring range. The narrow measuring range of GMI sensor was a significant disadvantage.

### 2.4. Results on Sensitivity of GSR Sensor

The effect of pulse current frequency on the sensitivity of GSR sensor is studied by changing the frequency on the GSR sensor with the length of 0.26 mm from 1 GHz to 3 GHz as shown in Figure 7a. The coil voltage increased with the increase of frequency following saturation over 3 GHz. The spins existing in the surface at the angular velocity ω (=2πf) and the high speed spin rotation excited by GHz pulse current makes the big coil voltage V (= −Δφ/Δt). The reason to take saturation over 3 GHz is because the actual frequency of the pulse current passing through the wire becomes lower than input frequency because of the strong eddy current. The sensitivity increases proportionally to the number of coils as shown in Figure 7b, where the number of coil turns change from 16 turns to 148 turns keeping their wire lengths of 0.96 mm. The increase of coil resistance and parasitic capacitance accomplished by the increase of number of coil turns do not affect present test result carried out under the present test conditions. The sensitivity increases proportionally to the effective permeability as shown in Figure 7c, where wires tested with intrinsic permeability of 1800 and the diameter of 10 μm have effective permeability of 150 and 460 controlled by wire length of 0.16 mm and 0.26 mm, respectively keeping the number of coil turns to 14. From this it is found that the sensitivity of GSR sensor is affected by pulse frequency, the detection type of falling or rising, the number of coil turns, and the effective permeability.

### 2.5. Results on Other Magnetic Properties

It is surprising that the rising detection of the GSR sensor makes no hysteresis or falling detection. GSR effect only detects spin rotation around the wire surface so that hysteresis does not occur. On the contrary, the GMI sensor shows a big hysteresis [1,34] in the case of rising detection because it detects axial magnetization. Rising detection is important for developing high ODR type GSR sensor of over 1 MHz. This means that GSR sensors potentially have a greater capability than GMI sensors. 

Figure 8 shows the result that σ-noise decreases to 40 μV under H = 7200 A/m when falling detection is carried out around the peak position of the coil voltage. This indicates that the magnetic noise of the GSR sensor is only 10 μV since the ASIC has its own noise of 30 μV. The frequency of pulse current takes the designated GHz frequency around peak position, but around the initial and ending time of pulse current it rises or falls slowly to take low frequency of KHz to MHz.

High frequency generates spin rotation accompanied by low noise. Low frequency induces domain wall movement to make big noise proportional to magnetic field strength.

Figure 9 shows the effects of tension treatment [2,33] at room temperature on the sine functional relationship between the coil voltage and the magnetic field. When the tension changes from 76 kg/mm^2^ to 10 kg/mm^2^, the sine function of 10 kg/mm^2^ shows distortion from the sine function of 76 kg/mm^2^. The reason for this is likely because the tension may enrich the circular spins on surface domain and compress the 90 degree domain wall inside the wire, so that the movements of magnetic walls are suppressed and bring forth the GSR effect to give the correct sine function dominantly. 

### 2.6. Summary of the Results

We observed the GSR effect based on the spin rotation of electron spins existing in the surface circular magnetic domain driven by GHz pulse current. The effect creates new features such as the development of great sensitivity by increasing coil voltage with pulse frequency and the extension of the range of linearity as well as the giving of non-hysteresis and low noise underlined by its sine function relationship between magnetic field and coil voltage. These features are explained by spin rotation not accompanied by magnetic wall movements.

## 3. Development of 3 Dimensional Photography Process for a Micro Coil

We developed a 3-dimensional photography process to produce a micro coil [3] and put it on the ASIC surface directly. The size of GSR sensor can be drastically downsized by one piece assembling with the element and ASIC as shown in Figure 10. The element is produced through the process shown in Figure 11, where a glass coated amorphous wire with a diameter of 10 μm has a composition of Co_50.7_Fe_8.1_B_13.3_Si_10.3_ and permeability of 1800. Figure 12 and Figure 13 show SEM photos on bottom coil pattern on the groove and top coil pattern on the convex respectively.

The first step is to form a resign film with the thickness of 10 μm and then make a groove with a width of 18 μm and depth of 7 μm on the film by RIE etching. Second step is to produce a bottom coil pattern with a coil pith of 5.5 μm. The 3-dimensional photolithography makes wire patterns on the convex-concave plane controlled by diffraction phenomenon between mask lattice and convex-concave plane in Figure 14. The coil pitch of 5.5 μm can be formed by the combination of mask lattice pitch of 5.5 μm and groove depth of 7 μm using the light wavelength of 700 nanometers (nm).

The third step is to set the amorphous wire along the groove using the wire alignment machine in Figure 15. This machine can improve the linearity of GSR sensor and align wires with the aliment interval of ± 1 μm accuracy (using the wire as a baseline for alignment), by applying tension between 50 kg/mm^2^ to 100 kg/mm^2^ on the wire with diameter of 10 μm. The fourth step is to mold the wire using an adhesive resist. The fifth step is to produce a wire coil pattern with a coil pith of 5.5 μm using 3-dimensional photolithography on the adhesive resist.

The above process can produce a micro coil with a coil pitch of 5.5 μm, coil diameter of 16 μm, wire length of 0.10 mm to 2 mm, and number of coil turns from 10 turns to 1000 turns on the ASIC surface. Essentially, GSR sensors can achieve a micro size while keeping the good performance of the sensors. 

## 4. Development of ASIC Type GSR Sensor for Various Applications

Various prototypes of GSR elements produced are shown in Figure 16. They are divided into one axis type with the length of 0.16 mm, 0.45 mm, and 0.99 mm, two axis type, and three axis type.

Some prototypes of GSR sensors suitable for specified applications [30,32] are produced by combining with these elements and the ASIC. The properties of these prototypes of GSR sensors are shown in Table 1. It is noted that the examinations have been carried out using one ASIC, which means that their performances are not optimized for all elements. This suggests that the GSR sensors have great potential for some specified applications. 

### 4.1. Automotive Use Application or Robot Industries

These applications require precise magnetic sensors of 16 to 18 bits that can equip a wide measuring range of over 80 G, and have high accuracy and sensitivity, good linearity, no hysteresis, low noise, low consumption, and wide bandwidth of over 500 KHz. Types of X11, X12, and X13 with the wire length of 0.16 mm can give the wide measuring range of over 80 G and good linearity of 0.1% FS, almost no hysteresis, low σ-noise of 2 mG to 6 mG, low current consumption of 0.4 mA, and 2.4 mG(240 nT)/LSB in condition of analog circuit bandwidth of 500 KHz and ODR of 5 KHz. In addition, it is well-known that amorphous wire type sensors equip strong reliability and temperature stability against outside atmosphere such as environment temperature, magnetic damage, and mechanical stress.

Previously, the GMI sensor had been expected to be the most promising sensor for automotive use, however it did not come to use because of its narrow measuring range of 12 G. The prototype of GSR sensor has already shown wide measuring range of over 80 G as well as high total performance, 100 times better than that of commercial ASIC type GMI sensor. Here the total performance is calculated by performance index of S/N ratio ×measuring range ×element size.

### 4.2. Small Size GSR Sensor for In-Body Use

GSR sensors can make very small size possible because its elements can be produced directly on the ASIC surface. The size of the GSR sensor can make the same size as the ASIC size of 1.2 mm × 1.2 mm × 0.1 mm that is used in this paper. This means that GSR sensor is promising for in body navigation use.

The magnetic devices with σ-noise of over 10 mG for in body navigation such as catheters, endoscope, and so on are used, but they have the poor positioning accuracy of 1–2 mm.

If types of X12 and X13 with the length of 0.45 mm and 0.90 mm respectively with σ-noise of 1 mG @ ODR of 5 KHz is used, it is expected that the positioning accuracy will improve to under 0.1 mm. These applications request long and thin shaped sensors that can be achieved with direct on-chip type of GSR, since this consists of long wires.

### 4.3. Compass for Smartphone and Mobile Computer

Types of XY and XYZ are operated to output five data such as X1, X2, Y1, Y2, and temperature at ODR of 1 KHz. Type XY for a 2D compass consists of two X-axis coils (X1 and X2) and two Y-axis coils (Y1 and Y2) to obtain the magnetic field at the center position by averaging to have a noise of 1.4 mG at ODR of 1 KHz and a range of 50 G. Type XYZ for a 3D compass consists of a 2D compass and permalloy parts to detect a Z-axis magnetic field. The sensitivity for the Z-axis magnetic field is adjusted by the height of the permalloy part. It is important to form a magnetic circuit by direct connection with the wire and the permalloy parts. It has noise of a 2.7 mG at ODR of 1 KHz and the range of 50 G.

Types of XY and XYZ are designed suitable for next generation compasses that request noise under 1 mG at ODR of 200 Hz and a measuring range of 24 G, compared to current specification of noise under 10 mG at ODR of 50 Hz and a range of 12 G. The new specification is about 20 times higher than the conventional one. Types XY and XYZ have not yet satisfied the new specification, but if ASIC performance or GSR element design is changed to make ODR from 1 KHz to 200 Hz and the measuring range 50 G to 24 G, the σ-noise will decrease from 1.4~ 2.7 to about 0.3 ~0.7 to satisfy the specifications needed for next generation compasses. These next generation compasses will have high speeds and accuracy so that it can calculate real time 3-dimensional attitude. That is, the next generation compass will be a magnetic Gyro-Compass with gyro functionality without the vibration gyro. This type of GSR sensor would be promising for use in devices such as smartphones, mobile computers, drones, robots, and goggles.

### 4.4. pT Sensor for Detecting Biomagnetism

The sensitivity of the GSR sensor can be improved through an increase in the number of coil turns produced by the increase in the length of the long wires or the four wires of GSR element. The GSR sensor is promising for these applications. However, a high-power electronic circuit with VDD of 5 V would be needed since the wire resistance of the long coil becomes more than 2 KΩ.

## 5. Conclusions

The GSR effect observed through excitement of GHz pulse current produced mainly four new features on the GSR sensor. One of these features is the increased sensitivity of the sensor, caused as a result of the increase in frequency of up to 3 GHz and coil turn numbers, and effective permeability. Another feature is the sensor’s extended range of linearity because of its sine functional relationship with the magnetic field and coil, expressed as V = V_0_・sin(πH/2Hm). The third feature is the low hysteresis of GSR sensor, being nearly zero, compared to the high hysteresis of GMI sensor. The fourth feature is the sensor’s ability to give minimal noise at detection timing. 

Considering the GHz frequency range used for the GSR sensor, we assume that the physical phenomena associated with the operations of the sensor are based on spin reduction and rotation of the magnetization.

We developed the production technology to produce a micro sized GSR sensor by directly forming micro coils on the ASIC surface. Some prototypes of ASIC type GSR sensor have been produced in consideration for applications such as automotive use, in body use, gyro–compass use, and medical use, as we have mentioned in the above sections. It is clear that GSR sensors have the potential to become the major magnetic sensor for many applications in the future.

## Figures and Tables

**Figure 1 sensors-20-01023-f001:**
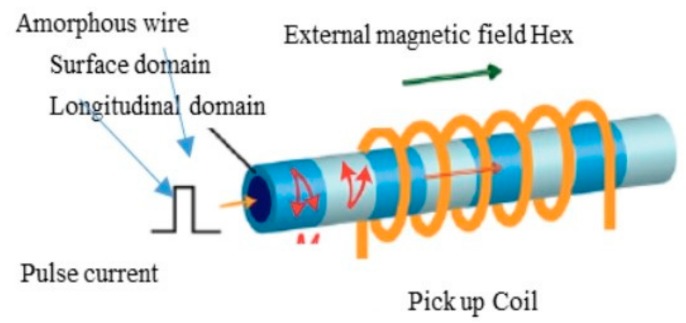
Principle of GigaHertz spin rotation (GSR) effect.

**Figure 2 sensors-20-01023-f002:**
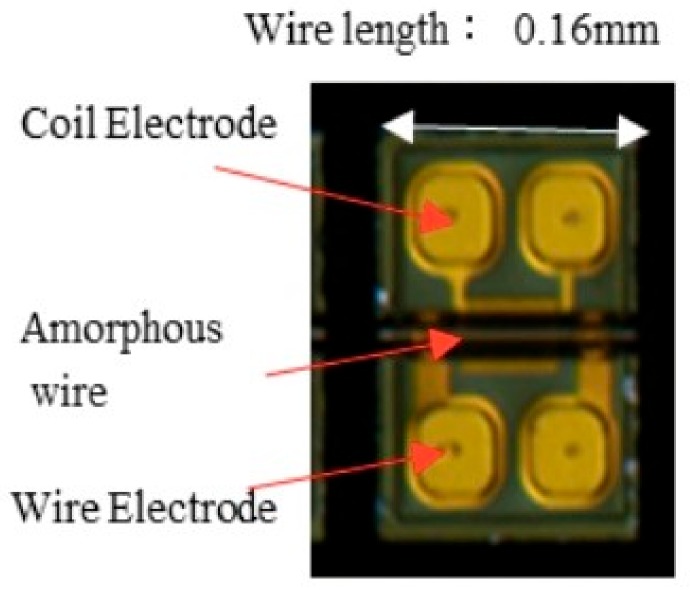
Structure of GSR element.

**Figure 3 sensors-20-01023-f003:**
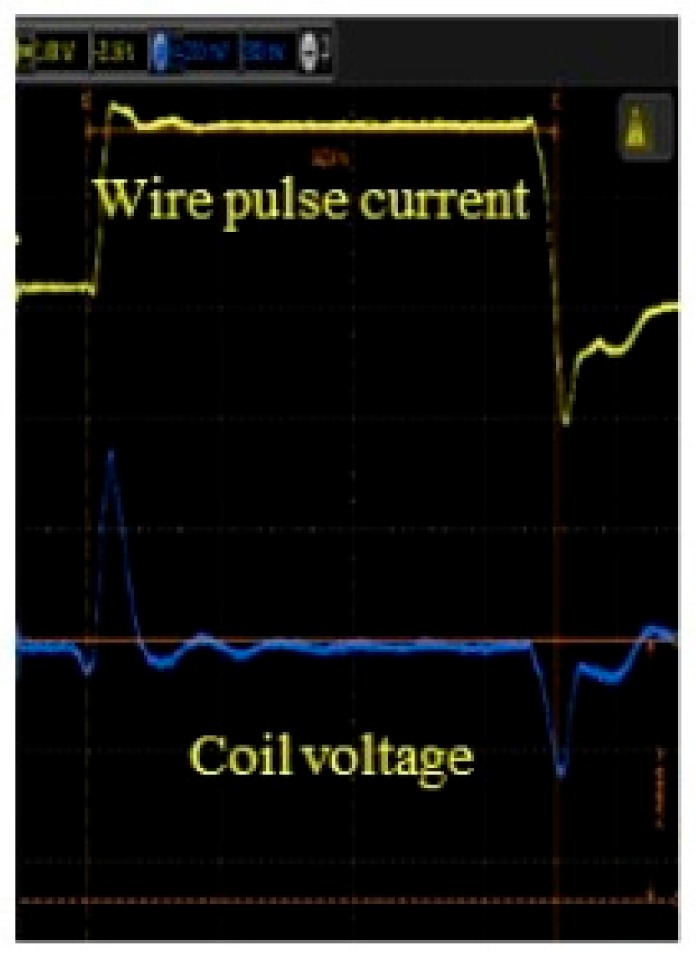
Observed coil voltage.

**Figure 4 sensors-20-01023-f004:**
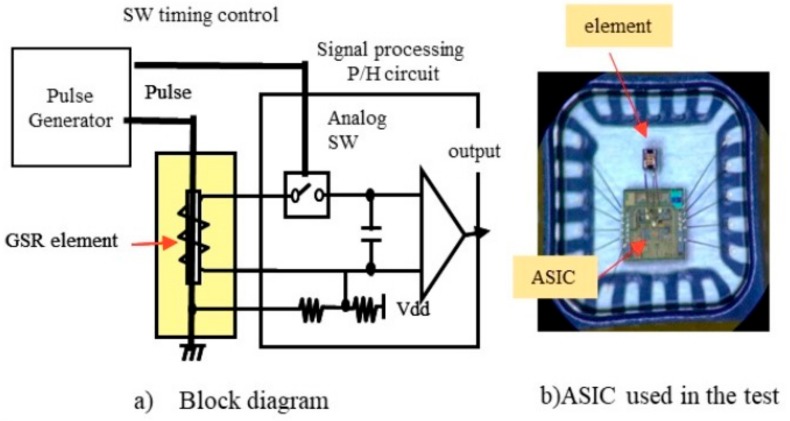
The circuit of GSR sensor.

**Figure 5 sensors-20-01023-f005:**
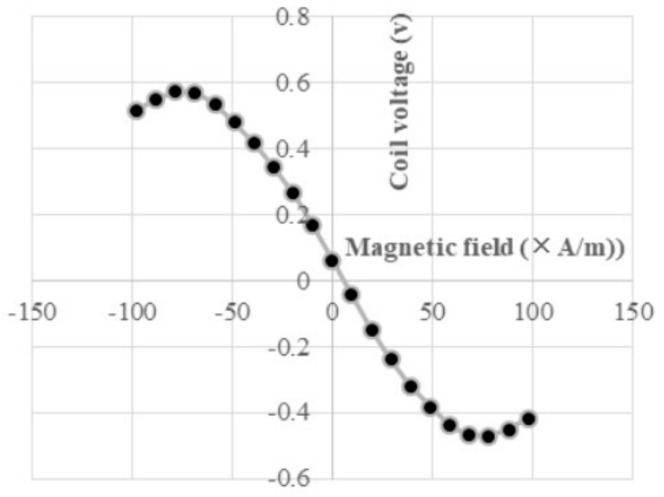
Coil voltage vs. magnetic field.

**Figure 6 sensors-20-01023-f006:**
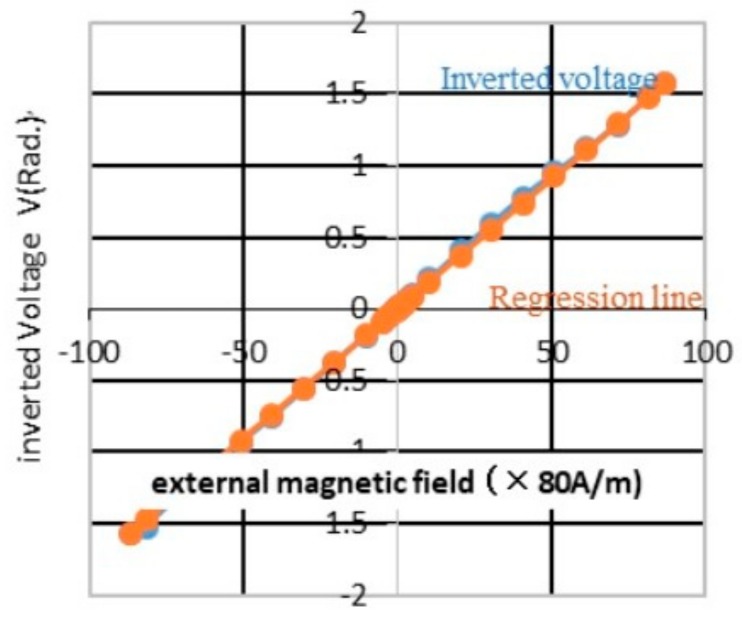
Inverted coil voltage vs. regression line.

**Figure 7 sensors-20-01023-f007:**
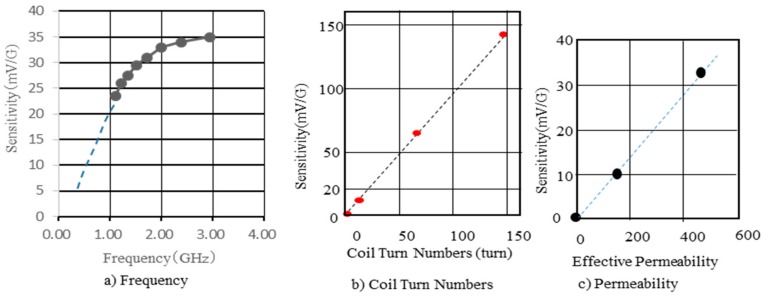
Effect of frequency, coil turn numbers, and effective permeability on sensitivity.

**Figure 8 sensors-20-01023-f008:**
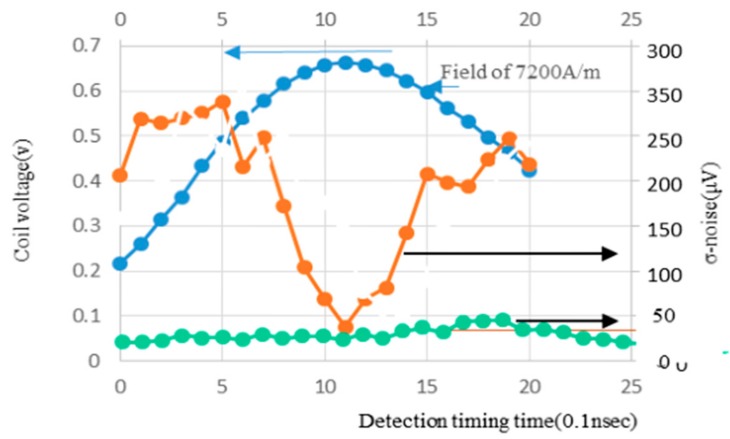
Coil voltage, noise (when field of 7200 A/m), noise (when Field of 0 A/m).

**Figure 9 sensors-20-01023-f009:**
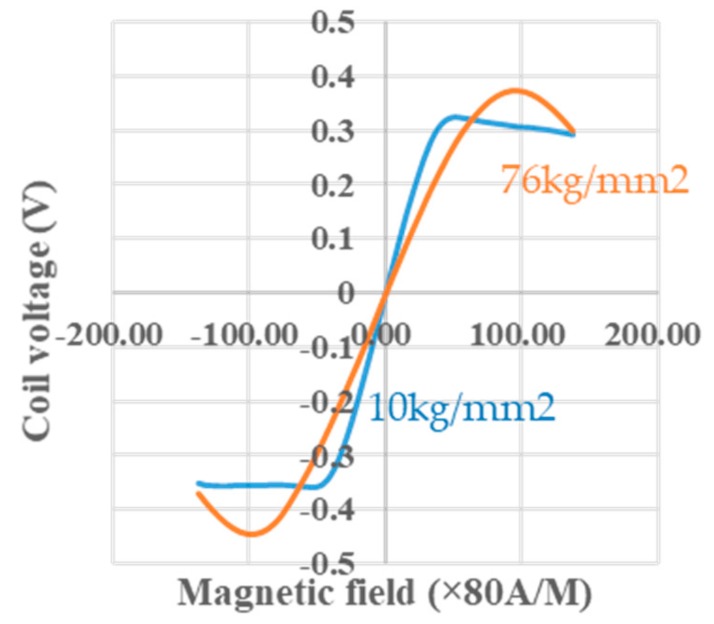
Effects of tension on coil voltage.

**Figure 10 sensors-20-01023-f010:**
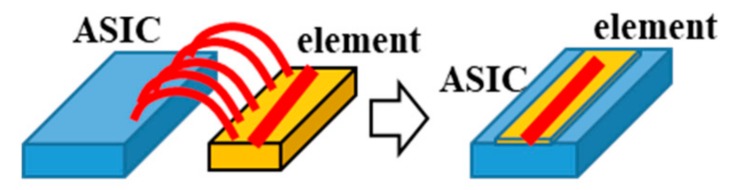
A wire bonded type GSR sensor and an on-ASIC type sensor.

**Figure 11 sensors-20-01023-f011:**
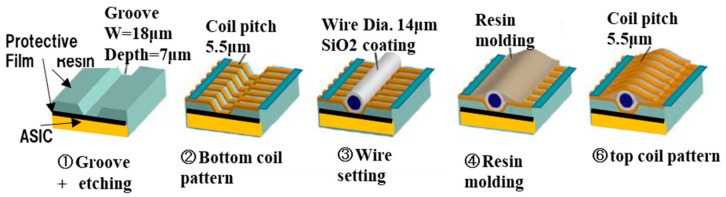
Production process to produce GSR element.

**Figure 12 sensors-20-01023-f012:**
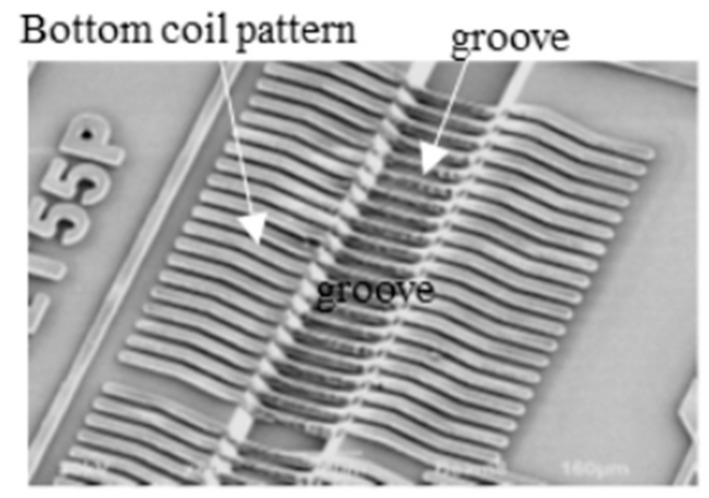
Bottom coil pattern on the groove.

**Figure 13 sensors-20-01023-f013:**
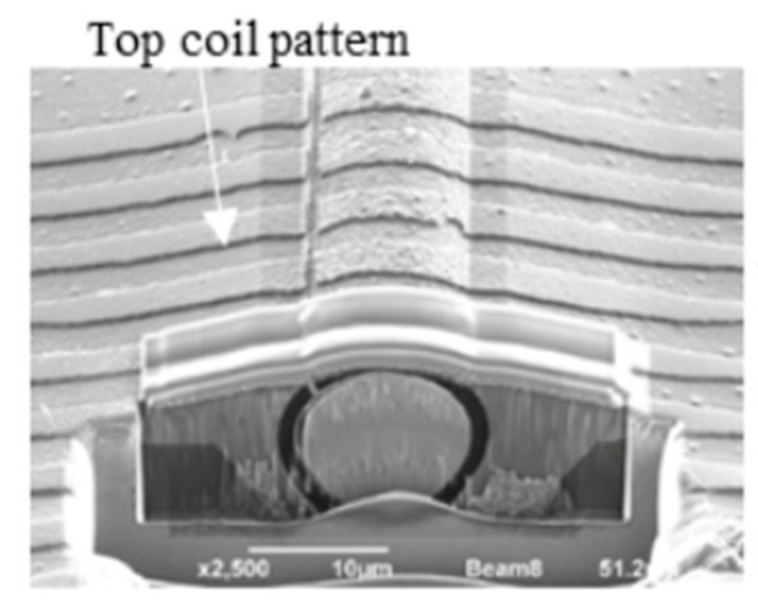
Top coil pattern on the convex.

**Figure 14 sensors-20-01023-f014:**
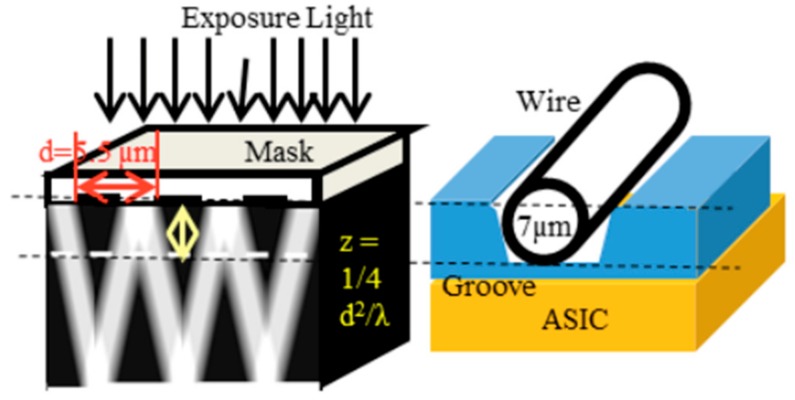
Diffraction phenomena by mask lattice.

**Figure 15 sensors-20-01023-f015:**
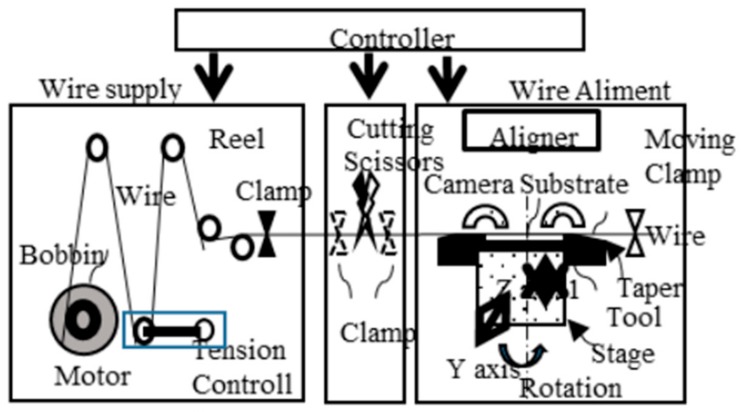
Wire aliment machine with the tension of 76 kg/mm^2^.

**Figure 16 sensors-20-01023-f016:**
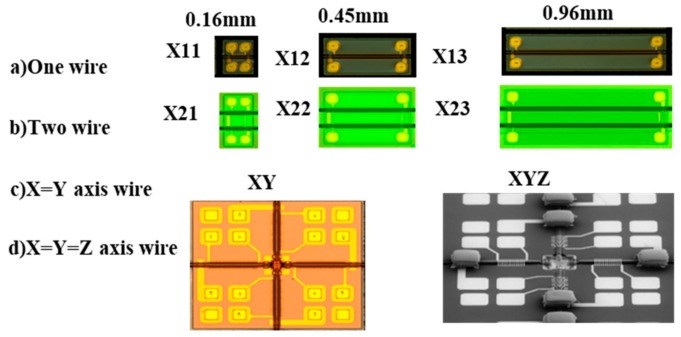
Various prototype GSR element.

**Table 1 sensors-20-01023-t001:** Performance of various type GSR sensors.

Types	Element Size Length (mm) × width (mm)	Resistance Wire/Coil	Coil Turn Numbers	Sensitivity	Noise	Noise	S/N Ratio	Measuring Range	Type of Future Application
		Ω	turn	Mv/G	µV	mG		A/m	
X11	0.16 × 0.23	3/80	14	10	60	6	167	6400	automotive (wide range)
X12	0.45 × 0.23	7/330	64	63	60	1	1050	2400	positioning (sensitivity)
X13	0.90 × 0.23	14/740	148	140	140	1	1000	Over 800	nT meter (high sensitivity)
X21	0.22 × 0.34	6/140	28	13	35	2.7	370	6400	automotive (wide range)
XY	0.26 × 0.3	6/160	32	30	70	2 (1.4)	430	4000	encoder (2D)
	(XY: 0.6 × 0.6)								
XYZ	0.26 × 0.3	6/80	14	16	60	3.8 (2.7)	270	4000	gyro compass (3D compass)
	(XYZ: 0.6 × 0.6)								
*MI	0.6 × 0.35	10/1	16	3.3	70	7	47	960	compass

* Commercial MI sensor.

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
