# Peer review of "The Development of ASIC Type GSR Sensor Driven by GHz Pulse Currentâ€"

_sensors, 2020, doi:10.3390/s20041023_

Round 1
Reviewer 1 Report
I recommend this paper to be published with some conditions
the authors must:
To extend the conclusions
To describe the problem clearly
They should make it clear what the purpose of the paper
Author Response
Point 1: To extend the conclusions
I have extended the conclusion to support it with the results from my research.
Point 2: They should make it clear what the purpose of the paper
In the introduction, I have included a short note to more clearly state the purpose and the outline of the paper.
Reviewer 2 Report
General comments:
The paper presents briefly all the stages of the production technology of micro-sized GSR sensor by directly forming micro coils on an ASIC surface. In the first part, the sensor is very well characterized and its properties are compared against similar FG- and GMI sensors. Also, some prototypes of ASIC type GSR sensor were presented for applications in automotive, human body use, gyro–compass and medicine. GSR sensor described here has a high degree of originality and an impressive potential to become very sensitive, non-hysteretic and low noise magnetic sensor; therefore, current article fits perfectly to the journal. However, English-language corrections and figures presentations will need VERY CAREFUL revision first. A large number of abbreviations need first to be introduced in whole words.
Please find bellow just some of the observations. They refer mainly to editing problems (lots), and not to the structure of the paper or to its technical content, which are very consistent.
Abstract:
None of the abbreviations are explained first in whole words; please introduce each abbreviation used even in the abstract
1.Introduction
41-42: please revise the phrase 64: please introduce whole words for abbreviation GSR, as first encountered here 65: please introduce whole words for abbreviation IOT, as first encountered here, etc...R 92 – ASIC,…(the same for each abbreviation which was not introduced before, there are many in the article) 72 – H is not explained2.Research on GSR Effect
R 113: instead of “;” please insert “:” Figs. 1 - 4 – it is adviced to increase the resolution of the images R 122: the wire resistances are: 3/8/4.5/13 ohms; is this order ok? R 153: it should be “from 960 V/m” R 155: “BH curve” – should be explained in full words first R 160: the title of the sub-section should appear below Figs. 5&6 R 165: Fig 7a is mentioned; but Fig. 7 is missing from the paper! Figs. 8 & 9 are not entirely explained: please insert legend in the figures to indicate different colored curves; the arrows (blue and black) are not explained; generally, on the Ox axis it is the variable, and on the Oy axis is the function; therefore, please provide clearer dependencies in the caption of the fig. 8; also, on Ox axis of Fig. 9 it is not the tension expressed in kg/mm2, as explained in the text. Fig. 9 needs a legend of the colored curves also.3.Development of ASIC Type GSR Sensor for Various Applications 237
R 251: 0.1%References
Reference [3] contains an editing error, at the end Refernce [30] contains repetitions.Author Response
I have revised the article based on your reviews, with the exception of the following points:
General comments:
The paper presents briefly all the stages of the production technology of micro-sized GSR sensor by directly forming micro coils on an ASIC surface. In the first part, the sensor is very well characterized and its properties are compared against similar FG- and GMI sensors. Also, some prototypes of ASIC type GSR sensor were presented for applications in automotive, human body use, gyro–compass and medicine. GSR sensor described here has a high degree of originality and an impressive potential to become very sensitive, non-hysteretic and low noise magnetic sensor; therefore, current article fits perfectly to the journal. However, English-language corrections and figures presentations will need VERY CAREFUL revision first. A large number of abbreviations need first to be introduced in whole words.
Reply: I have introduced all the whole words for necessary abbreviations.
Please find bellow just some of the observations. They refer mainly to editing problems (lots), and not to the structure of the paper or to its technical content, which are very consistent
Reply: As I will explain below, I have revised the editing problems that you have observed.
Abstract:
None of the abbreviations are explained first in whole words; please introduce each abbreviation used even in the abstract
Reply: I explained the whole words for all and each of the abbreviations that appear for the first time in the abstract.
1.Introduction
41-42: please revise the phrase 64: please introduce whole words for abbreviation GSR, as first encountered here 65: please introduce whole words for abbreviation IOT, as first encountered here, etc...R 92 – ASIC,…(the same for each abbreviation which was not introduced before, there are many in the article) 72 – H is not explained
Reply:
For R 41-42, I changed my phrasing from saying “It is worth mentioning that” to simplifying this to just “Notably”. For R 64, and 92, I did not introduced the whole words for the abbreviations of GSR and ASIC, since they were already introduced in the abstract. However for R 65, I did introduce the whole word for the abbreviation IoT. As for R 72, I noticed my mistake that I previously had not included the H with μ as “μH”.2.Research on GSR Effect
R 113: instead of “;” please insert “:” Figs. 1 - 4 – it is adviced to increase the resolution of the images R 122: the wire resistances are: 3/8/4.5/13 ohms; is this order ok? R 153: it should be “from 960 V/m” R 155: “BH curve” – should be explained in full words first R 160: the title of the sub-section should appear below Figs. 5&6 R 165: Fig 7a is mentioned; but Fig. 7 is missing from the paper! Figs. 8 & 9 are not entirely explained: please insert legend in the figures to indicate different colored curves; the arrows (blue and black) are not explained; generally, on the Ox axis it is the variable, and on the Oy axis is the function; therefore, please provide clearer dependencies in the caption of the fig. 8; also, on Ox axis of Fig. 9 it is not the tension expressed in kg/mm2, as explained in the text. Fig. 9 needs a legend of the colored curves also.
Reply:
For R 113, 122, 153, and 155, I have revised as advised by the reviewer, such as inserting “:” instead of “:”, for changing the order of data for resistance, changing the phrase to “from 960 A/m”, and explaining the full words for “BH curve”. For Figures 1-4, I have increased the resolution of these images and moved Figures 5 & 6 as advised. Although the reviewer has mentioned that 7a was mentioned in the paper but 7 is missing, I believe we have included all instances of Fig. 7, from 7a to 7d. For Figures 8 & 9, I have newly inserted a legend below each figures to indicate the different colored curves. The Blue and Black arrows used in Figure 8 were originally supposed to indicate that the blue curve as pointed by blue arrow is a Coil voltage curve, and the orange and green curves as pointed by black arrows are noise (μV) curves. We can revise the legend for Figure 8 more if that would help clarify the data.
3.Development of ASIC Type GSR Sensor for Various Applications 237
R 251: 0.1%
Reply: For R 251, I have edited “o.1%” as “0.1%” as advised.
References
Reference [3] contains an editing error, at the end Refernce [30] contains repetitions.
Reply: For [3] and [30], I have fixed the editing error and repetitions as specified.
Reviewer 3 Report
The authors are encouraged to improve the quality and resolution of several figures almost all affected by blurring. Furthermore, they should follow the jorunal's template and the correct caption of the figures. They should also better define some referenced acronyms. Finally, in the conclusions section they should better explain what are the advantages of the proposed solution and what are the consequent applications.
Author Response
Based on your reviews, I have made revisions on my article, such as improving image qualities, and defining acronyms.
The authors are encouraged to improve the quality and resolution of several figures almost all affected by blurring.
Reply: I have improved the resolutions of figures that were affected by blurring.
Furthermore, they should follow the jorunal's template and the correct caption of the figures.
Reply: In the Instruction for Authors link, it is only mentioned that all figures must be labeled with their corresponding number of appearance, title, and caption. I believe I have labeled all my figures with correct title, number, and caption. If there are any specific errors that you observed, please let us know.
They should also better define some referenced acronyms.
Reply: I revised to define referenced acronyms in their first appearance as advised.
Finally, in the conclusions section they should better explain what are the advantages of the proposed solution and what are the consequent applications.
Reply1: Although not revised in the resubmitted paper, I revised the conclusion in the final resubmission paper, with more explanation for the consequent applications and the advantages of the proposed solution.
Reply2:For explaining the advantages of GSR sesnor, I added the medical applicatons such as Magneto- cardiogram (MCG) and Magneto-encephalography (MEG) in the section of pT senor for Detecting Biomagnetism( IV. D)
Thank you for spending your time in reviewing my article, and I greatly appreciate your cooperation.
Round 2
Reviewer 3 Report
The paper is now ok.